# Effect of an Antioxidant Based on Red Beetroot Extract on the Abiotic Stability of Polylactide and Polycaprolactone

**DOI:** 10.3390/molecules26175190

**Published:** 2021-08-27

**Authors:** Petra Drohsler, Jaroslav Cisar, Tomas Sopik, Vladimir Sedlarik, Martina Pummerova

**Affiliations:** Centre of Polymer Systems, University Institute, Tomas Bata University in Zlín, Tr. T. Bati 5678, 76001 Zlín, Czech Republic; pvalkova@utb.cz (P.D.); jcisar@utb.cz (J.C.); sopik@utb.cz (T.S.); sedlarik@utb.cz (V.S.)

**Keywords:** polylactide, polycaprolactone, beetroot, antioxidant, biodegradable polymer, abiotic stability, degradation

## Abstract

This study investigated the effect of natural antioxidants inherent to beetroot (*Beta vulgaris* var. *Vulgaris*) on the ageing of environmentally friendly plastics. Certain properties were examined in this context, comprising thermal, mechanical, and morphological properties. A visual evaluation of relevant changes in the given polymers (polylactide and polycaprolactone) was conducted during an ageing test in a UV chamber (45 °C, 70% humidity) for 720 h. The films were prepared by a casting process, in which samples with the extract of beetroot were additionally incorporated in a common filler (bentonite), this serving as a carrier for the extract. The results showed the effect of the incorporated antioxidant, which was added to stabilize the biodegradable films. Its efficiency during the ageing test in the polymers tended to exceed or be comparable to that of the reference sample.

## 1. Introduction

Plastic has many advantages as a material, such as its minimum weight, transparency, and durability, that distinguish it from others. Its production in recent years has risen consequentially, especially though the manufacture of disposable items. There are related drawbacks, though, most notably the consumption of nonrenewable resources (petroleum), from which many types of plastic are made, and a large quantity of subsequent waste that damages the natural environment. As an ever greater number of problems have been identified with petroleum-based plastic, greater interest is evident in alternative polymeric materials made from renewable sources, especially for packaging purposes [1,2,3]. Considerable attention has been paid to “biodegradable polymers,” which are characterized as capable of degradation through nonenzymatic processes such as chemical hydrolysis and mainly by the enzymatic activity of microorganisms that utilize bacteria, fungi, and algae [4,5]. Industrial composting equipment are applicable for carrying out this task, wherein the controlled degradation of organic material (waste from the kitchen or garden) takes place at an elevated humidity and temperature and under regular aeration. The resulting humus is commonly applied as a natural fertilizer [3]. Materials that also show the potential to undergo such a composting process are biodegradable polymers, i.e., polyesters with hydrolytically labile ester bonds, that affect the rate of degradation, and other properties have been reported in the scientific literature by Kucharczyk et al. [6]. The group includes aliphatic polyesters such as polycaprolactone (PCL) and polylactide (PLA), the properties of which classify them as suitable for a wide range of applications. Biodegradable polyesters have been employed in the development of numerous products, examples being biocomposites, nonwoven materials, medical devices, and packaging items [6,7,8].

The PLA is a polymer derived from lactic acid monomers produced by the fermentation of by-products, particularly carbohydrate-rich substances such as corn, potatoes, and sugar [9]. The direct condensation polymerization reaction of PLA is made possible by the presence of the hydroxyl and carboxyl groups in lactic acid [7,10,11]. PLA usually takes the form of a transparent thermoplastic, very similar to polystyrene. The glass transition temperature (T_g_) of the polylactide is influenced by the crystalline structure of the polymer during thermal processing, where the temperature of the atactic homopolymer is 60 °C, and that of the crystalline homopolymer is up to 70 °C. The melting point (T_m_) of the polymer, which lies in the range of 160‒190 °C, is dependent on the molecular weight and crystalline phase [12,13,14].

PCL is a linear semicrystalline polyester composed of hexanoate repeating units. As a polylactide, polycaprolactone is chiefly reliant on the molecular weight and the degree of crystallinity. During PCL synthesis, the ions and metals catalyzed by the ring-opening polymerization (ROP) of the cyclic monomer ε-caprolactone are predominant. PCL is partially crystalline, with a glass transition temperature (T_g_) of −60 °C and a melting point (T_m_) of 59–64 °C. The hydrolysis of PLA and PCL polymers occurs through ester bonds. PCL contains five hydrophobic -CH_2_ groups in its repeating units, which causes it to degrade slowest out of all polyesters (3‒4 years). This makes the polymer PCL suitable for applications where long degradation times are required [15,16,17]. However, these materials are subject to premature and undesirable degradation processes, which limit their applicability. For example, abiotic factors exert effects under conditions of storage; such factors include light, temperature, UV radiation, and, notably, oxygen, which acts synergistically with humidity. This affects the physical and chemical integrity, thereby altering the mechanical, thermal, barrier, and optical properties of the material. Stabilizers and antioxidants (AOs) are used as a countermeasure to hinder abiotic processes [18,19]. The function of AOs is to eliminate the reactive forms of oxygen and nitrogen through the provision of their hydrogen from the hydroxyl group [20].

Numerous synthetic AOs are available commercially, yet few suitable natural AOs exist for packaging purposes, especially for active packaging [21,22,23]. Examples of widely used synthetics include butylated hydroxyanisole (BHA), butylated hydroxytoluene (BHT), and tertiary butylated hydroquinone (TBHQ) [24,25,26]. Research on natural AOs have centered around phenolics, polyphenolics, bioactive pigments, and tannin substances commonly obtained from plant extracts, such as coffee, cocoa [27,28], tea [29], grapes [30], citrus fruits [31], spices and medicinal herbs [32], and agricultural waste products (berries fruits or tuberous vegetables) [33].

Chief examples of bioactive pigments include betalains, which are present in high concentrations in beetroot (BR) (*Beta vulgaris).* Betalains are often utilized in foodstuffs as a natural water-soluble colorant. There are two forms of betalain in BR, betacyanins (red-violet pigments) and betaxanthins (yellow-orange pigments) [34,35]. The antioxidant effect of betalains is much higher than those of many flavonoids, ascorbic acid, and tocopherols. It has also been shown that betalamic acid, the basic structure of all betalains, is therefore able to donate two electrons to an oxidizing agent [36,37].

However, betalains show sensitivity to stability influenced by factors that include degradation by heat, light, enzymes, and oxygen. During storage and processing, betalains can degrade, causing changes in color and antioxidant effect [34,38].

Many studies have investigated natural AOs for use in packaging applications to extend the shelf life of foodstuffs and/or stabilize the given polymeric material [39]. The current trend is to add AOs into environmentally friendly materials, such as polysaccharide edible films for packaging purposes [40,41,42]. Nevertheless, a deficiency exists in the description of the properties of natural AOs, especially betalains, in biodegradable polymers prepared from commonplace PLA and PCL.

In the future, the ecofriendly biodegradable polymers will be promising packaging materials. For this reason, the presented work aimed to improve PLA and PCL polymers using available, cheap, and natural AO obtained from beetroot (*Beta vulgaris)*. The chemical characterization of extracted AO was examined. Generally, natural AOs should be stabilized; in this research, the incorporation of AO on a common inert carrier (bentonite) was studied. Moreover, the PLA and PCL polymer composites were blended with stabilized AO and subsequently characterized in terms of thermal, chemical, and structural stability under abiotic conditions, especially UV irradiation and temperature.

## 2. Results and Discussion

### 2.1. Characterization of Beetroot Extract (BRE)

The polyphenol content of the extract was quantified by Folin–Ciocalteu reagent (FCR) and such an assay was performed for the entire BR bulb. The results were evaluated according to the calibration series prepared, through application of the gallic acid standard. The various sections of the BR constituted significant sources of polyphenol. Indeed, the entirety of the BRE contained 93 mg (expressed as mg of gallic acid equivalent per 100 g of BRE). The content of polyphenolic substances in the BRE, as prepared in the ethanol solution (70%), reached similar values to those in the literature. This BRE was seen to remain stable under conditions of storage at low temperature (4–8 °C) in darkness for at least one month, with no significant change in polyphenol content [43,44].

The antioxidant activity of the BRE was evaluated by the 2,2-diphenyl-1-picrylhydrazyl (DPPH), a scavenging method widely deployed for evaluating antioxidant activity for relatively short durations, compared to other methods. The main attribute of AO is the neutralization of free radicals by donating an electron or hydrogen atom. In particular, polyphenols more often act as direct radical scavengers of the lipid peroxidation chain reaction (chain-breakers). The cessation of the reaction chain and the formation of a stable radical occurs upon the supply of an electron from the chain-breaker to the free radical, which it neutralizes [45,46]. The reducing ability of the DPPH radicals was determined by diminishing its absorbance at 517 nm, as induced by the AOs. The subsequent measurements revealed that the antioxidant activity of the BRE corresponded to 114 mg per the equivalent amount of 100 g of ascorbic acid. The value for antioxidant activity, herein converted to the amount of ascorbic acid, is influenced by various aspects such as the type of BR storage, conditions, the environment of its cultivation, and the technique of extract preparation. This explains the wide variance reported in other publications, i.e., from 20 to 170 mg (expressed as mg of ascorbic acid equivalent per 100 g of BRE) [47,48,49].

Figure 1 shows a representative total ion chromatogram (TIC) for the BRE from mass spectrometry in positive ion mode. On integration, the TIC exhibited several peaks evaluated as ms/ms fragmentation obtained arising through collision-induced dissociation. The peak at 2.98 min, a [M + H]^+^ molecular peak of *m*/*z* 549.1336, presented a fragment of *m*/*z* 387.0809 and was consistent with previous findings on the fragmentation of neobetanin (Figure 2A) published in references [50,51]. The molecular ion at 2.49 min had a value for *m*/*z* of 551.1509, which, on dissociation, yielded fragments of *m*/*z* 389.0976 (Figure 2B) typical for betanin or isobetanin (2.61 min) [50,51,52]. In coelution with betanin and isobetanin, the following were identified: 2′-*O*-glucosyl-betanin (t_R_ = 2.52 min; [M + H]^+^ = 713.2034) and 2′-*O*-glucosyl-isobetanin (t_R_ = 2.63 min; [M + H]^+^ = 340.1132) [51]. Tentative identification was also performed on feruloylglucose at 3.55 min ([M + H]^+^ = 357.1175), 5,5′,6,6′-tetrahydroxy-3,3′-biindolyl at 3.74 min ([M + H]^+^ = 297.0874), betavulgarin at 5.35 min ([M + H]^+^ = 313.0704), and cochliophilin A at 6.23 min ([M + H]^+^ = 283.0604) [50,51].

### 2.2. Thermal Analysis

The applicability of polyphenol-containing BRE as a natural source of AOs added to polymers is crucially dictated by their thermal stability. The thermal properties of the prepared samples with natural AOs from BR were analyzed by DSC and TGA and were compared with the neat polymers (PLA and PCL) and these polymers with the BE carrier alone. A significant effect of ageing on the values of T_g_ for the PLA samples was observed after 720 h in the UV chamber (see Table 1). As a result of accelerated ageing, a clearly noticeable decrease in T_g_ values was observed for the neat PLA sample, potentially caused by two phenomena. First, low-molecular-weight compounds may have formed new macromolecules in the process of degradation, giving rise to plasticizing properties and lowering the glass transition temperature of the polymeric materials. Secondly, the macromolecules could have been shortened by breakage of the polymeric main chain. It is known that short macromolecules are characterized by low glass transition temperatures. As for the ageing study of semicrystalline PCL, it was not possible to compare the glass transition of the material (−60 °C) technically. Analysis of the PCL samples revealed that data on T_m_ could be evaluated, with neat PCL and PCL-BE samples showing two melting peaks (Table 2). The absolute values for the cold crystallization and enthalpy of melting of these samples are virtually identical, as ageing PCL often exhibits double-melting behavior, i.e., a stable melting structure at high temperatures and a less stable melting structure at lower temperatures [53].

Visible changes were only observed in the thermal stability values of T_peak_ in the presence of oxygen, where samples with the carrier (bentonite) and the carrier with the AO demonstrated more rapid thermal degradation than samples comprising the pure polymers did. This phenomenon occurs to a lesser extent in the sample before PLA aging, especially in PLA-BE (Table 1, Figure 3). The primary difference between the polymers was the intrinsic enthalpy ∆H of the PCL-BRE samples (1157 J/g), which is by far the lowest. Table 2 and Figure 4 and reveal that the thermal degradation of the PCL-BRE samples proceeded slowly, more so than for the samples under comparison. In the case of PCL-BRE, the effect of the anti-degradant as an interrupter of the chain autooxidation reaction caused the formation of free radicals from the macromolecular chain to cease. Generally, the thermal decomposition of both polymers can be affected by several external abiotic factors, such as temperature, moisture content, or the presence of oxygen. Another important factor is that free radicals are responsible for the thermal degradation of these polymers. The thermal degradation of polymers is considered an inevitable effect under normal melt processing conditions. However, degradation can be prevented or slowed down by an inorganic filler or AOs [50,51].

The curves for the TGA, PLA, and PCL composites indicate the influence of not only bentonite, but also the AO on the stability of both materials after solar ageing. The PLA curves in Figure 5 show that the initial release of moisture and other volatiles occurred first (a weight loss in the region below the thermal degradation). The onset of PLA-UV degradation was already apparent at 354 °C, appearing for PLA-BE-UV at 343 °C and PLA-BRE-UV at 347 °C. The main degradation peaks were visible at 340 °C for PLA-UV, at 331 °C for PLA-BE-UV, and at 335 °C for PLA-BRE-UV. Degradation was at a slower rate for PLA-BRE-UV manifested by an initially slower loss in mass (Table 3). The TGA curves in Figure 6 describe the degradation process of the PCL materials. The effect of bentonite was obvious, reducing the rate of degradation for PCL-BE-UV to 367 °C and PCL-BRE-UV to 360 °C, whereas PCL-UV was evident at 335 °C [54,55].

### 2.3. Mechanical Properties

The mechanical properties of the PLA and PCL composites were monitored prior to and following the artificial ageing test, and compared with the reference samples shown in Figure 7 and Figure 8. The first of the parameters was monitored by Young’s modulus (Figure 7). This parameter increased, especially in the case of aging PLA samples, where the material became rigid and more brittle, caused mainly by the cleavage of the chains of macromolecules and the crosslinking. The maximum stress point (Figure 8) of the materials was also observed. A slight increase in this parameter was observed for composites with bentonite, which acted as a nucleation agent. Another important parameter concerned elongation at break (Table 4). For the PLA and PCL-based systems before ageing, adding a natural AO led to a rise in elastic modulus. The slight increase in elongation at break in all AO systems (PLA-BRE and PCL-BRE) in comparison with neat PLA and PCL and also PLA-BE and PCL-BE samples could be attributed to a plasticizing effect initiated by stabilizing molecules; indeed, it is known that low-molecular-weight molecules dispersed in polymeric matrices can increase the free volume of the system and reduce friction between macromolecules. PCL-BRE demonstrated a higher elongation of 1062% before aging, dropping to 375% after aging, and the reference sample showed a reduction from 1006% to 14%. The composite PLA-BRE prior to ageing exhibited 363%, and 9% afterward, whereas the relevant reference dropped from 357% to 24%. These types of chemical modifications of the polymeric materials, which occur in the photo-oxidation process due to decomposition reactions, are detrimental to the mechanical properties of the materials.

According to the results illustrated in Table 4 as per the ageing factor (A_f_), the reference sample PLA appeared to be the most stable under extreme conditions in the climatic chamber. PCL samples supplemented with the AO from BRE also exhibited a suitable ageing factor (A_f_). This study showed a low effect of natural AO from BR, i.e., BRE could reduce the amount of synthetic AO in degradable polymer formulations, especially PCL-based ones. Hydrolytic degradation of the chains in PLA occurs primarily on the surface and preferably in amorphous regions [55,56].

### 2.4. Colour Measurement

It is known that additives applied to change the color of polymeric materials in food packaging can affect the overall appearance and, thus, the consumer. In particular, light in the UV region can trigger photooxidation processes, with the potential rapid loss in quality or deterioration of the packaged foodstuffs as a consequence, making such a step undesirable [57], although unwanted alteration may occur in the original color during transport and storage. It is important to examine the effect of these additives on the individual properties of the materials. The results presented herein reveal the effect of current natural additions of bentonite and extract on the biopolymers (Figure 9 and Figure 10). The altered color of all the films was immediately noticeable, in that they turned a shade of yellow. In the case of the PLA films, the change was most pronounced in the sample containing only bentonite. The color change index (∆E) for this was 29, i.e., the same value for the neat PCL sample (Figure 9). In contrast, the lowest values were evident for the samples with BRE, and the color change index equaled 17 in both cases. During the ageing test, a significant alteration occurred in chromium (C_ab_) in the biopolymer samples only supplemented with bentonite. The whiteness index (W_i_) increased in all the samples due to a change in the transparency of the materials, and a slight haze appeared during ageing. For these reasons, incorporating such natural dyes in the given polymers would reveal the degree of degradation and could be used as indicators (Figure 10).

### 2.5. Fourier-Transform Infrared Spectroscopy (FT-IR)

The effect of ageing on the PLA and PCL films was determined by FTIR, thereby gauging the extent of alteration in chemical structure. The subject of this measurement was to monitor changes in structures due to the added BRE. Figure 11 displays spectra for the PLA before and after exposure to UV. The ester bonds (C(=O)-O-C) of PLA located in the bands 1088 cm^−1^ and 1183 cm^−1^ represented sensitive groups mainly (C-O-C) in the process of solar aging. The bands observed at 1383 cm^−1^ and 1455 cm^−1^ were caused by CH_3_ asymmetric and symmetric deformation. Other characteristic PLA peaks at 1756 cm^−1^ corresponded to (C=O) carbonyl groups. The intensity of the aforementioned sensitive bands decreased through UV radiation. To conclude, the samples of neat PLA demonstrated a significant decrease in the characteristic bands, mainly due to photodegradation or thermal degradation of the polymer structure. Certain peaks for PLA-UV in the region of 1654 cm^−1^ were also discerned, indicating the formation of anhydride groups upon degradation [58,59].

The main peaks characteristic of the PCL material (Figure 12) were found primarily in the regions of 2865 cm^−1^ and 2945 cm^−1^, representing symmetric and asymmetric bending vibrations of the methylene unit. Another was a peak at 1724 cm^−1^ pertaining to the stretching of the carbonyl of the ester group. It is possible that C-O-C bands corresponded to the peaks at 1187 cm^−1^ and 1241 cm^−1^. The PCL samples underwent degradation upon exposure to the UV chamber and temperature, revealed through heightened peaks for the carbonyl groups, thereby suggesting the formation of radicals in the PCL macromolecules. In the case of PCL-UV, the main peak broadened at 1724 cm^−1^ [60,61].

Although no structural chemical changes were observed in both matrices after the addition of BE and BRE (before the UV chamber), there were minor changes in individual cases after exposure to the UV chamber.

However, all the samples experienced partial degradation, additionally affected by the higher temperature in the UV chamber; this acted on the material for a relatively long time; therefore, it is not possible to directly deduce any unambiguous effect by the BRE.

### 2.6. Scanning Electron Microscopy (SEM)

SEM images of fracture surfaces of PLA and PCL samples (Figure 13A–F and Figure 14A–F) detail changes before and after exposure in the UV chamber, at 1000× magnification, but also sample surfaces after the UV chamber (G‒I), at 270× magnification. SEM images made it possible to monitor the surface changes caused by the UV chamber environment. It was also possible to observe stratification and homogeneity of the added filler at the fractures of the samples. The degradation process was evident after four weeks in the UV chamber, with cracks, cavities, and peeling fragments visible in both materials [60,61]. However, PLA samples showed lower changes after exposure in the UV chamber than PCL samples did. Degradation was best observed on the surface of PCL, which was significantly degraded by visible cavities after exposure. Samples of PLA and PCL containing BRE also showed signs of degradation, although these manifestations were mild and affected only a thin surface layer.

## 3. Materials and Methods

### 3.1. Materials and Reagents

Commercial PLA (2003D) from NatureWorks (Minnetonka, MN, USA) and PCL from Sigma-Aldrich (Saint Louis, MO, USA) were utilized in the experiments. Fresh red beetroot (BR) was obtained from a local farm (Ostrozska Nova Ves, Czech Republic). Bentonite was sourced from local wine producers at the particle size >20 µm. Ethanol (99.8%) was purchased from BC-Chemservis (Roznov pod Radhostem, Czech Republic). Acetone (99.88%) was obtained from Chromservis (Prague, Czech Republic). Folin & Ciocalteu’s phenol reagent (2 M), gallic acid (97.5%), ascorbic acid (reagent grade), sodium carbonate (anhydrous, Emsure™ACS, ISO, Reag. Ph Eur), and folin-, 1,1-diphenyl-2-(2,4,6 trinitrophenyl) hydrazyl came from Merck (Darmstadt, Germany) and Sigma-Aldrich (Saint Louis, MO, USA). Tetrahydrofuran was supplied by Carl Roth Rotisolv^®^ HPLC (Karlsruhe, Germany) and chloroform was bought from Penta (Prague, Czech Republic). Water and acetonitrile Chromasolv™ Plus solvents for HPLC and LC-MS were purchased from Honeywell GmhB (Seelze, Germany) and formic acid for LC-MS LiChropur™ was obtained from Sigma Aldrich (Saint Louis, MO, USA).

### 3.2. Preparation of BRE and Films

#### 3.2.1. Preparation of BRE and Incorporation into the Carrier

The authors lyophilized fresh BR, having employed a CoolSafe 110-4 PRO freeze dryer in its preparation (Lynge, Denmark).

There is a simple and efficient preparation of the extract, which is based on the following preparation: 250 mg of homogenized and freeze-dried BR with 10 mL of extraction solvent (70% ethanol) was added into a centrifuge tube, and this homogenate was then stirred on a Vortex device (IKA^®^ MS 3 basic, Staufen, Germany) for 2 min at 500 rpm. The resultant extract was centrifuged (Thermo Scientific™, Heraeus Multifuge X1R, Osterode, Germany) for 15 min, at 9000 rpm. The supernatant (7 g) was collected with a Pasteur pipette and applied to the bentonite (BE) (5 g) as a carrier. After mixing, the samples were lyophilized. The additive immobilized in this way is more suitable for handling and advantageous for technological processing.

#### 3.2.2. Preparation of the Films

Cast films were prepared by dissolving the polymer and adding bentonite with the incorporated BRE (5% *w*/*w*, based on the weight of the polymer) in chloroform. The solution was stirred at room temperature for 8 h to completely distribute the antioxidant in the polymer matrix. The polymer solution was then poured into Petri dishes of 140 mm diameter. Evaporation of the chloroform took place in a fume hood in darkness for 24 h at 25 °C. The subsequent films were dried in a vacuum oven (Memmert VO400, Frankfurt, Germany) at 25 °C for 5 h to remove residual solvent and sealed in a refrigerator subsequently. The eventual thickness of each film was 0.35 ± 0.05 mm. Neat PLA and PCL films and composites containing bentonite (BE 5% *w*/*w*) were prepared as reference samples for the important comparison of samples with films containing BRE. The procedure for preparing the polymer composites is shown in Figure 15, while the composition and names of the tested formulations are given in Table 5.

### 3.3. Process for Accelerated Ageing

Accelerated ageing was performed in a climatic chamber (QUV UV tester, Westlake, OH, USA) at 45 °C in temperature and 70% relative humidity under UV irradiation for 720 h. The experiment was designed in accordance with EN ISO 4892-3 [62]. The chamber was fitted with 8 fluorescent lamps (UVB 313) of 313 nm in wavelength. The samples were placed directly under the lamps (perpendicular to the UV source), which, after the accelerated ageing test, were denoted with suffix-UV to the names given in Table 5.

### 3.4. Determination of Antioxidant Activity and Polyphenol Content of the BRE

#### 3.4.1. Determination of Total Polyphenol Content of the BRE (The Folin–Ciocalteu Method)

Determining polyphenol content in the BRE involved application of the Folin–Ciocalteu method. The principle behind it is that the reduction of FCR contains a mixture of phosphotungstic acid and phosphomolybdic acid, which is reduced to a mixture of blue tungsten oxide(s) and molybdenum by oxidizing the phenols in the sample. The blue hue demonstrates maximum light absorption at the wavelength range of 750–760 nm, the intensity of this absorption being directly proportional to the total amount of phenolic compounds present. The resulting value is declared as the equivalent amount of gallic acid.

The supernatant of BRE at 0.1 mL in amount was added into 1 mL of freshly prepared 10% FCR. The mixture was allowed to equilibrate for 5 min in darkness and then treated with 1 mL of 10% sodium carbonate solution. After incubation at room temperature for 15 min, the absorbance of the mixture was read at 750 nm, using an appropriate solvent as a blank. Results were expressed as mg equivalents of gallic acid per 100 g of BR.

#### 3.4.2. Determination of Total Antioxidant Activity by 2,2-Diphenyl-1-picrylhydrazyl (DPPH)

The antioxidant activity of BRE was determined by a basic method based on the scavenging of free radicals using a DPPH. The principle of this method is the reaction of AOs (polyphenols) with the stable DPPH radical, which reduces the radical to form DPPH-H (2,2-diphenyl-1-picrylhydrazine). The reaction was observed by spectrophotometry at the wavelength of 515 nm, which corresponds to the maximum absorbance of the radical due to intense change in its color.

Then, 0.1 mL of BRE was mixed with 1 mL of 0.1 M of acetate buffer (pH 5.5) and 1.9 mL of 0.2 mM of DPPH. The sample was kept at room temperature in darkness for 60 min. The blank sample consisted of 1 mL of 0.1 M of acetate buffer (pH 5.5), 1.9 mL of 0.2 mM of DPPH, and 1.9 mL of 70% ethanol formulated for 60 min in the darkness at room temperature. The results were expressed as mg equivalents of ascorbic acid per 100 g of BR.

#### 3.4.3. HPLC-ESI-MS/MS Analysis

HPLC analysis of the beetroot extract was performed on a 1260 Infinity LC system (Agilent Technologies, Santa Clara, CA, USA). Chromatographic separation of the samples was carried out on a ZORBAX Extend C18 column (50 mm × 2.1 mm, 1.8 µm) (Agilent Technologies, Santa Clara, CA, USA) at a flow rate of 0.3 mL·min^−1^ that was maintained at 35 °C. The mobile phase consisted of 0.1% formic acid in HPLC-grade water (eluent A) and acetonitrile (eluent B). The following gradient was applied: 0.0‒1.0 min, hold at 10% B; 1.0–2.0 min, linear gradient from 10% to 50% B; 2.0–6.5 min, hold at 50% B; 6.5–10.0 min, linear gradient from 50% to 90% B; 10.0–17.0 min, hold at 90% B; the post-run lasted 3 min. The total running time was 17 min for each sample; the sample injection volume equaled 2 µL.

Detection was performed by a quadrupole time-of-flight mass spectrometer (6530 Q-TOF, Agilent Technologies, Santa Clara, CA, USA), employing a source of electrospray ionization (ESI) set to positive ion mode. The mass spectrometer operated under the following parameters: capillary voltage 3500 V, nebulizer pressure 40 psi, drying gas 8 L·min^−1^, and gas temperature 300 °C. Mass spectra were acquired over the *m*/*z* 100–1500 range at a scan rate of 3 scan·s^−1^. Accurate mass measurements were obtained by means of a calibration solution, involving the use of internal reference masses (purine (C_5_H_4_N_4_) at *m*/*z* 121.050873, and HP-0921 (hexakis-(1H,1H,3H-tetrafluoropentoxy)-phosphazene; C_18_H_18_O_6_N_3_P_3_F_24_ at *m*/*z* 922.009798). Data were recorded and processed in MassHunter software v.B.05.01 (Agilent Technologies, Santa Clara, CA, USA).

### 3.5. Characterization Techniques

#### 3.5.1. Thermal Properties

Thermal analyses were carried out to determine the oxidizing capacity of the polymer films with BRE, the techniques comprising differential scanning calorimetry (DSC) and thermogravimetry (TGA). The DSC tests were conducted to determine the glass transition temperatures (T_g_), cold crystallization temperature (T_cc_), melting temperature (T_m_), and enthalpy change (∆H_m_) of all the materials, which took place on a Mettler Toledo DSC1 STARe System (Schwerzenbach, Switzerland) under a nitrogen atmosphere (at the flow rate of 50 mL·min^−1^). The samples (~8 mg) were sealed in an aluminum pan and heated from −20 to 300 °C, and then cooled to 20 °C and reheated to 300 °C at the same cooling—heating rates.

Further DSC measurements were performed through determination of the thermal stability of the samples, conducted in the presence of an oxygen atmosphere at temperatures from 20 to 400 °C and a heating rate of 20 mL·min^−1^. Temperatures from the DSC curves are referred to as initial degradation temperatures (T_onset_), peak temperatures (T_peak_), and internal enthalpy (∆H).

Samples for TGA measurement were prepared at 8 mg in weight and investigated on a TA Q500 thermogravimetric analyzer (TA Instruments, Wilmington, DE, USA) and analyzed by a TA Universal Analyzer 2000 version 4.5A (TA Instruments—Waters LLC, Wilmington, DE, USA), which was set to a heating rate of 10 K·min^−1^ from 0 to 800 °C under a nitrogen atmosphere. Data on the temperature at the maximum rate of weight loss of samples (T_max_) were collected.

#### 3.5.2. Mechanical Properties

Mechanical tests were performed to characterize the stability of the samples in terms of material strength when ageing. Tensile strength testing was carried out on a universal tester (M350-5 CT Materials Testing Machine, Testometric, Lancashire, UK), under modified conditions based on EN ISO 527-3:2018 [63]. Prior to this, the specimens had been conditioned according to the ISO standard at 23 °C and 50% humidity for 24 h. Tensile testing was conducted at a constant rate of uniaxial deformation of 50 mm·min^−1^ until the samples ruptured, and this was carried out on least eight specimens from each group. The ageing factor (A_f_) was calculated concurrently for the samples based on Equation (1) [64]:A_f_ = (σ·ε)_before aging_/(σ·ε)_after aging_,(1)
where A_f_ is the ageing coefficient (-), σ is the tensile strength (MPa), and ε is the elongation at break (%).

#### 3.5.3. Color Measurement

Optical examination of the samples by color measurement was performed before and after the ageing test with a portable spectrophotometer (Lovibond RT850i colorimeter, Tintometer Ltd., Amesbury, UK). The CIELab scale was applied to measure lightness, *L**, and the chromaticity parameters, *a** (red–green) and *b** (yellow–blue). Color difference (ΔE), the whiteness index (W_i_), and chroma (C_ab_) values were calculated according to Equations (2)–(4). These parameters permit the estimation of the change in color of the polymer after incorporating the natural AOs into the matrix of the polyester blend [65].
(2)$$\Delta E=\sqrt{\left(\Delta a^{2}\right)+\left(\Delta b^{2}\right)+\left(\Delta L^{2}\right)}$$
(3)$$W_{i\;}=100-\sqrt{a^{2}+b^{2}+\left(100-L^{2}\right)}$$
(4)$$C_{ab}=\sqrt{a^{2}+b^{2}}$$

#### 3.5.4. Fourier-Transform Infrared Spectroscopy (FTIR)

Fourier-transform infrared-attenuated total reflectance (FTIR-ATR) was employed to obtain spectra for all the polymer films (Nicolet iS5, Thermo Fisher Scientific, Waltham, MA, USA). The unit was fitted with a Ge crystal to determine the structural alteration in chemical bonds under ageing conditions; it was set to the range of 600–4000 cm^−1^ and the resolution of 4 and 64 scans, with analysis in OMNIC software (Thermo Fisher Scientific, Waltham, MA, USA).

#### 3.5.5. Morphology

The cryo-fracture surfaces and the surfaces of the samples were gauged in a high vacuum environment at a test operating voltage of 10 kV, by scanning electron microscopy with a Phenom Pro desktop scanning electron microscope (SEM) with a BSE detector (Phenom-World B.V., Eindhoven, The Netherlands).

## 4. Conclusions

A great level of interest has been shown in using natural resources that are environmentally friendly, from which materials can be created as an alternative to nonbiodegradable conventional plastics. The research herein focused on devising a simple method for preparing biodegradable polymers, namely polylactide (PLA) and polycaprolactone (PCL) filled with a natural additive that possessed antioxidant and coloring effects. Furthermore, bentonite was used as a commonly available additive to polymers and, at the same time, served as a suitable carrier for the AO component. During testing, the influence of the AO additive was demonstrated, manifested through a diminished change in color, suggesting a potential function as an indicator of age-related degradation in the material. Determining the thermal stability of the samples by DSC analysis revealed a decrease in thermal properties. In the case of TGA, an effect of the antioxidant was discerned after the solar ageing test, comprising elevated temperatures during the gradual degradation of the material, as detailed in Table 3. Mechanical tests also detected the effect of bentonite on the brittleness of the material, whereby samples without the filler demonstrated high values for Young’s modulus after ageing; for neat PLA and PCL, the increase equalled 90% and 50%, respectively. However, bentonite reduced the final strength of the specimens during stretching. Elongation at break indicated a possible antioxidant effect in the PCL sample, which had a value of 375% compared to the others with ca. 10%. The findings presented suggest that additives from natural sources could be applied in packaging materials as an alternative option. However, further research must be carried out to find ways to stabilize this potential AO in the polymer matrix in order to reap its full benefits.

## Figures and Tables

**Figure 1 molecules-26-05190-f001:**
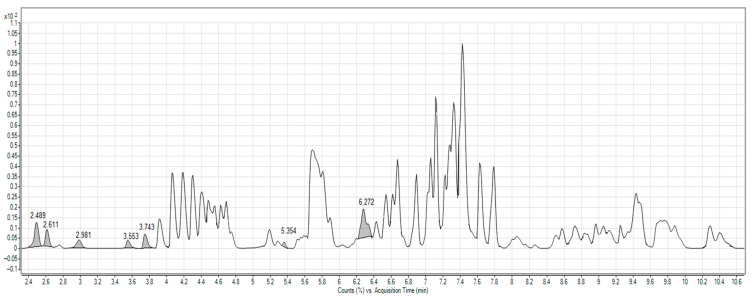
Representative TIC of diluted BRE; retention times were attributed by mass spectrometry analysis.

**Figure 2 molecules-26-05190-f002:**
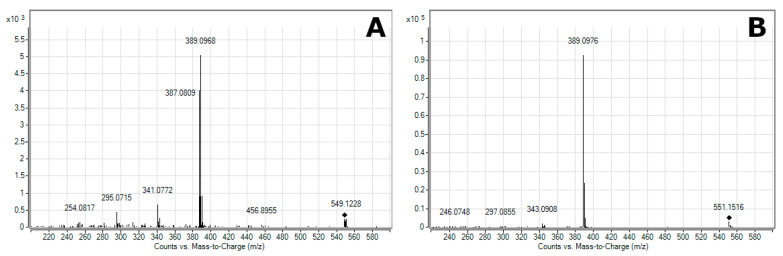
Positive electrospray tandem mass spectra for (**A**) neobetanin, and (**B**) betanin or isobetanin. The daughter ion of *m*/*z* 387.0809 was obtained by fragmentation of the parent ion of *m*/*z* 549.1228 of neobetanin. The daughter ion of *m*/*z* 389.0976 corresponds to protonated aglycone, and was obtained by fragmentation of the parent ion of *m*/*z* 551.1516 of betanin or isobetanin. The collision energy was 10 eV and 30 eV for (**A**) and (**B**), respectively.

**Figure 3 molecules-26-05190-f003:**
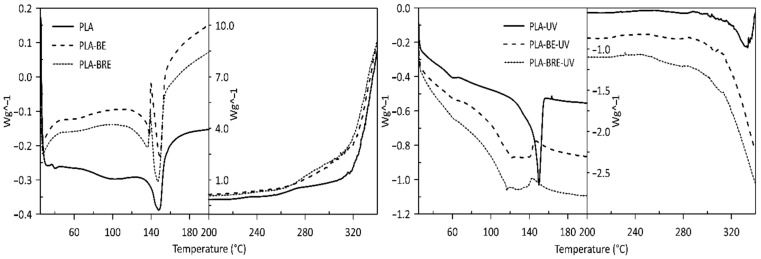
DSC degradation test for the PLA samples under an oxygen atmosphere.

**Figure 4 molecules-26-05190-f004:**
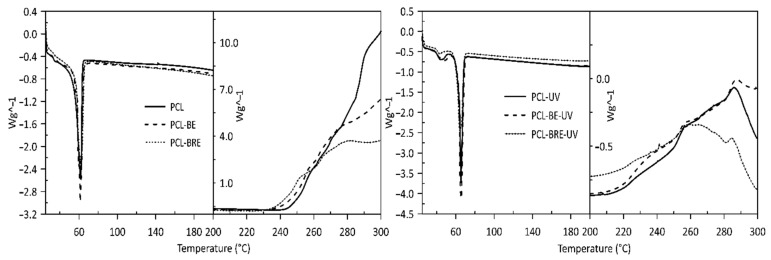
DSC degradation test for the PCL samples under an oxygen atmosphere.

**Figure 5 molecules-26-05190-f005:**
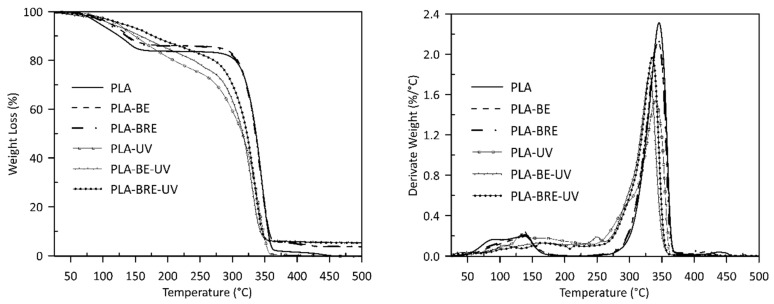
Thermogravimetric response of PLA bentonite with AO alongside the reference samples.

**Figure 6 molecules-26-05190-f006:**
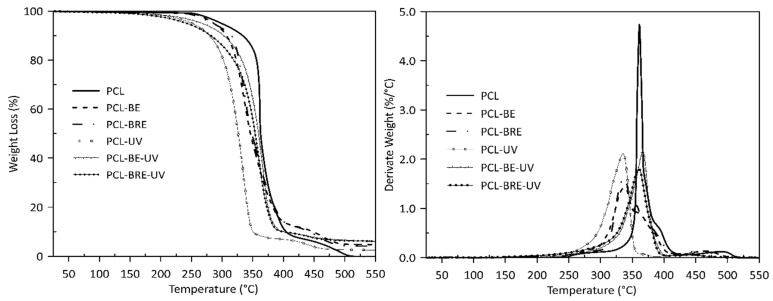
Thermogravimetric response of PCL bentonite with AO alongside the reference samples.

**Figure 7 molecules-26-05190-f007:**
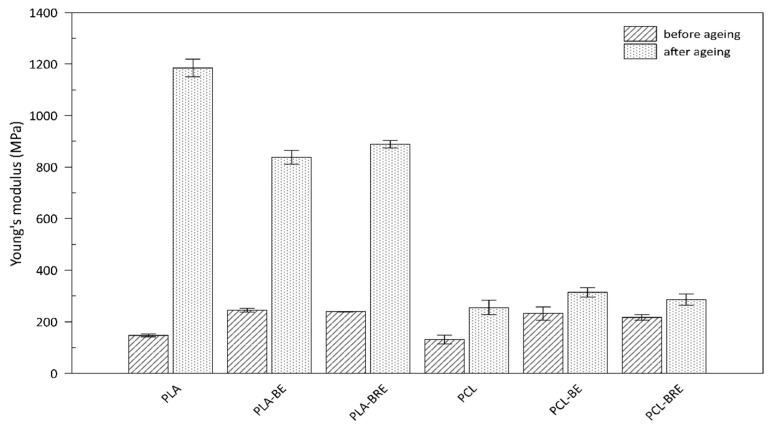
Mechanical properties of the PLA and PCL samples as per Young’s modulus.

**Figure 8 molecules-26-05190-f008:**
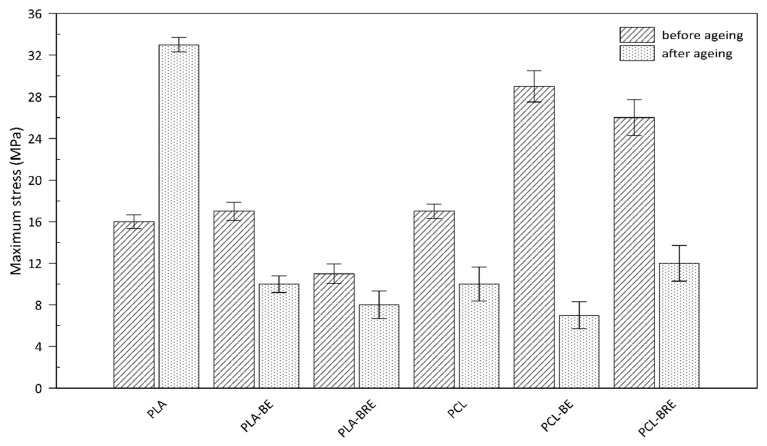
Mechanical properties of the PLA and PCL samples as per maximum stress.

**Figure 9 molecules-26-05190-f009:**
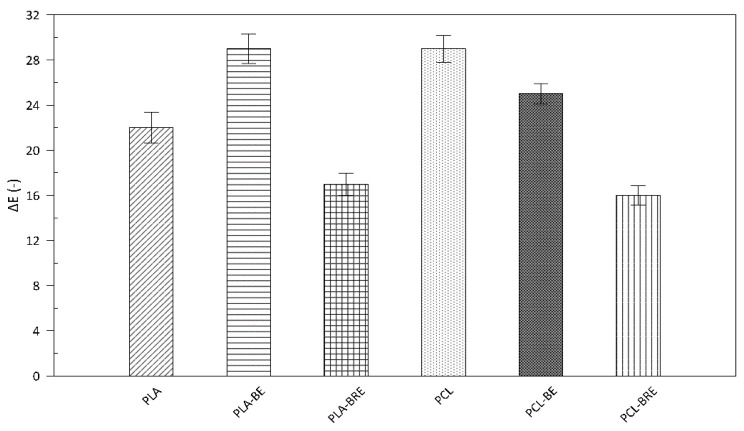
Color change (∆E) in the PLA and PCL composites with bentonite and natural AOs, compared with the reference of pure PLA and PCL; calculations were performed according to Equation (2).

**Figure 10 molecules-26-05190-f010:**
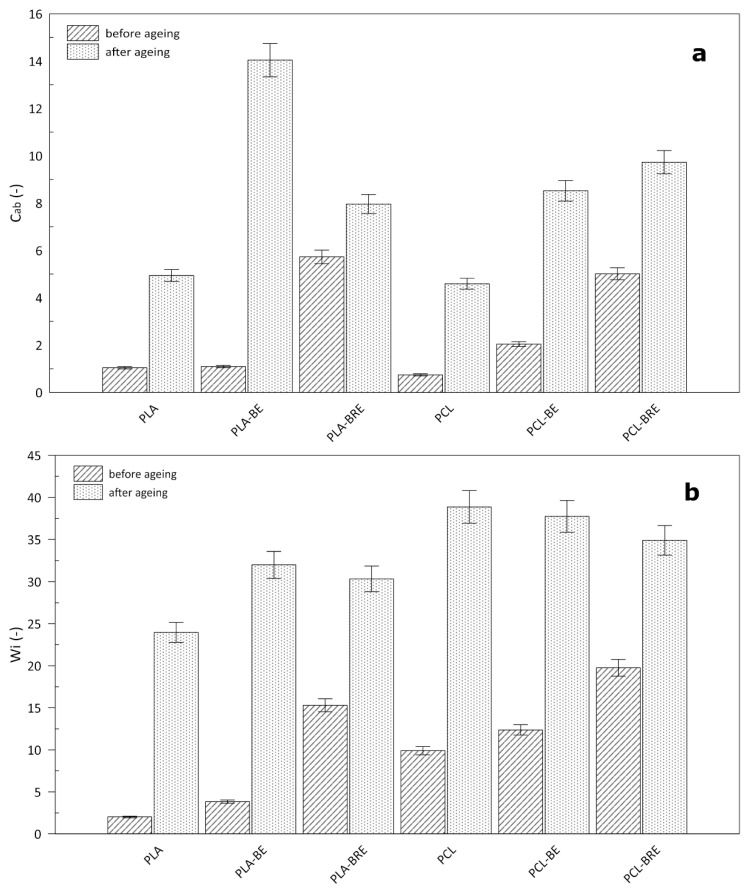
Impact of solar ageing on (**a**) chroma (C_ab_) and (**b**) the whiteness index (W_i_); calculations were performed according to Equations (3) and (4).

**Figure 11 molecules-26-05190-f011:**
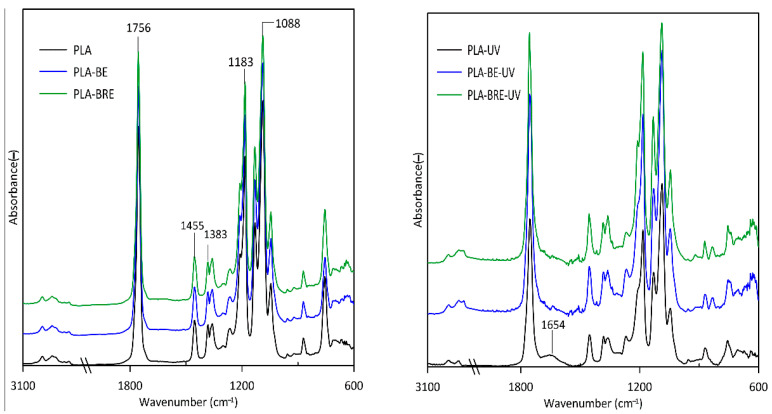
FTIR spectra for the PLA and PLA composite films prior to and following exposure in a UV chamber.

**Figure 12 molecules-26-05190-f012:**
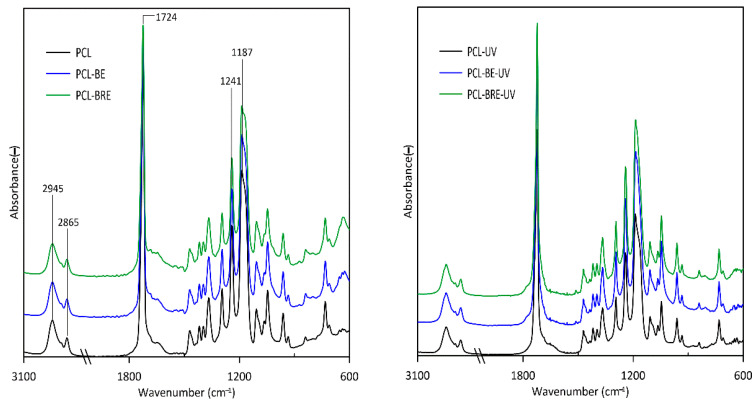
FTIR spectra for the PCL and PCL composite films prior to and following exposure in a UV chamber.

**Figure 13 molecules-26-05190-f013:**
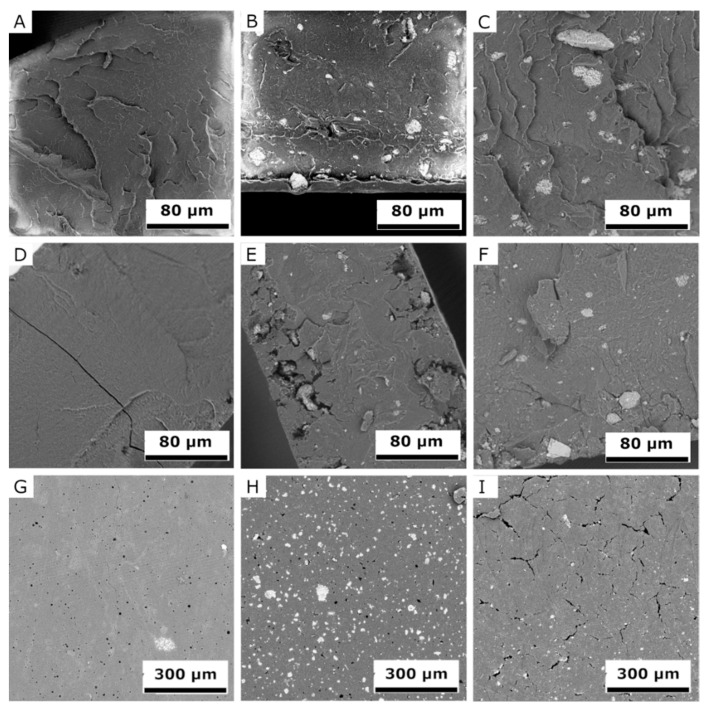
SEM images detailing change in the PLA samples prior to and following exposure in a UV chamber; the fracture surfaces of samples (**A**) PLA, (**B**) PLA-BE, (**C**) PLA-BRE, (**D**) PLA-UV, (**E**) PLA-BE-UV, and (**F**) PLA-BRE-UV; and the surfaces of the samples (**G**) PLA-UV, (**H**) PLA-BE-UV, and (**I**) PLA-BRE-UV.

**Figure 14 molecules-26-05190-f014:**
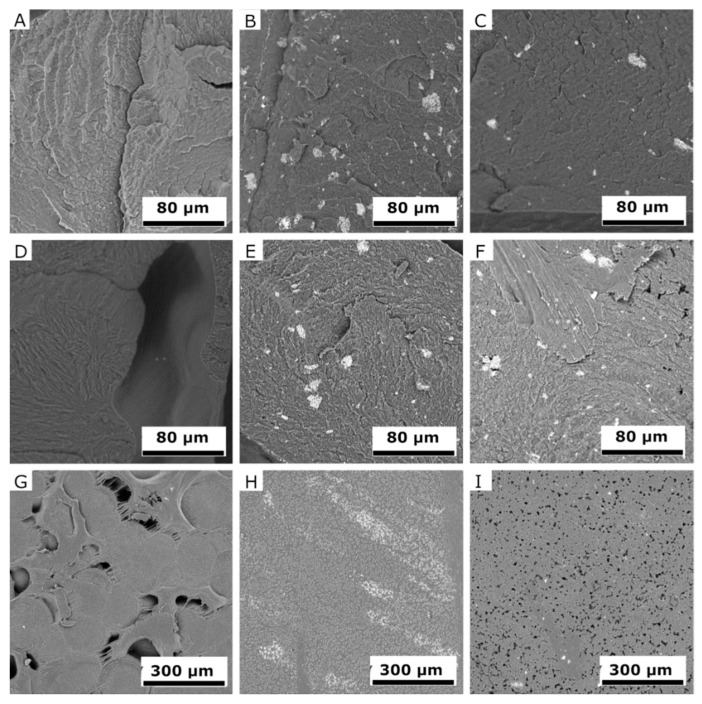
SEM images detailing change in the PCL samples prior to and following exposure in a UV chamber: the fracture surfaces of samples (**A**) PCL, (**B**) PCL-BE, (**C**) PCL-BRE, (**D**) PCL-UV, (**E**) PCL-BE-UV, and (**F**) PCL-BRE-UV; and the surfaces of samples (**G**) PCL-UV, (**H**) PCL-BE-UV, and (**I**) PCL-BRE-UV.

**Figure 15 molecules-26-05190-f015:**
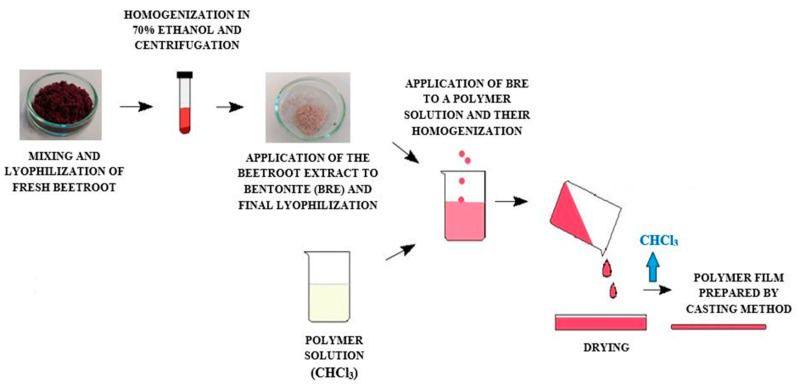
Preparation method of the samples.

**Table 1 molecules-26-05190-t001:** DSC data for the neat PLA and PLA composites with bentonite and AO before and after the *UV ageing* test.

Sample	T_g_ [°C]	T_m_ [°C]	∆H_m_ [J/g]	T_onset_ [°C]	T_peak_ [°C]	∆H [J/g]
PLA	53	146	36	254	355	1805
PLA-BE	57	143	39	267	349	1564
PLA-BRE	57	145	38	265	347	1746
PLA-UV	49	151	42	212	-	-
PLA-BE-UV	53	133	37	216	-	-
PLA-BRE-UV	57	135	39	218	-	-

**Table 2 molecules-26-05190-t002:** DSC data for the neat PCL and PCL composites with bentonite and AO before and after the *UV ageing* test.

Sample	T_m1_ [°C]	T_m2_ [°C]	T_cc_ [°C]	∆H_cc_ [J/g]	∆H_m_ [J/g]	T_onset_ [°C]	T_peak_ [°C]	∆H [J/g]
PCL	65	-	34	3	88	245	312	3168
PCL-BE	65	-	34	3	86	240	314	2254
PCL-BRE	66	-	35	3	90	236	281	1157
PCL-UV	66	39	26	3	92	215	286	254
PCL-BE-UV	64	38	36	3	99	216	288	307
PCL-BRE-UV	-	64	30	1	93	208	265	143

**Table 3 molecules-26-05190-t003:** Summary of TGA curves for the PLA, PCL, and composites.

Samples	T_onset_ (°C)	T_10_ (°C)	T_50_ (°C)	T_90_ (°C)	Mass Loss (%)
PLA	359	123	337	355	100
PLA-BE	358	140	336	357	95
PLA-BRE	357	136	335	356	95
PLA-UV	354	149	315	348	100
PLA-BE-UV	343	156	316	342	95
PLA-BRE-UV	347	175	322	346	95
PCL	368	333	363	350	100
PCL-BE	378	307	346	449	95
PCL-BRE	378	315	348	443	95
PCL-UV	350	278	326	350	99
PCL-BE-UV	381	302	359	397	95
PCL-BRE-UV	379	281	354	398	95

**Table 4 molecules-26-05190-t004:** Comparison of the mechanical properties of the PLA and PCL composites prior to and following ageing; the ageing factor (A_f_) was calculated according to Equation (1).

Samples	Before Solar Ageing		After Solar Ageing		A_f_ (‒)
	σ (MPa)	ε (%)	σ (MPa)	ε (%)	
PLA	15 ± 1	354 ± 18	16 ± 2	24 ± 6	0.07 ± 0.01
PLA-BE	16 ± 2	301 ± 25	6 ± 2	5 ± 1	0.01 ± 0.00
PLA-BRE	10 ± 1	363 ± 14	1 ± 1	9 ± 1	0.00 ± 0.00
PCL	17 ± 1	1006 ± 9	8 ± 3	14 ± 6	0.01 ± 0.01
PCL-BE	22 ± 2	1008 ± 6	6 ± 3	8 ± 3	0.00 ± 0.01
PCL-BRE	22 ± 2	1062 ± 8	11± 3	375 ± 42	0.15 ± 0.00

**Table 5 molecules-26-05190-t005:** Composition of the prepared samples.

Name	Composition
PLA	Neat PLA
PLA-BE	PLA + 5% *w*/*w* bentonite
PLA-BRE	PLA + 5% *w*/*w* bentonite with beetroot extract
PCL	Neat PCL
PCL-BE	PCL + 5% *w*/*w* bentonite
PCL-BRE	PCL + 5% *w*/*w* bentonite with beetroot extract

## Data Availability

Not applicable.

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
