# Peer review of "Effect of an Antioxidant Based on Red Beetroot Extract on the Abiotic Stability of Polylactide and Polycaprolactone"

_molecules, 2021, doi:10.3390/molecules26175190_

Round 1

Reviewer 1 Report

A main concern regarding the proposed work is that the PLA and PCL used in the study were from commercial sources and, normally, commercial polymers already contain antioxidant. If the authors did not remove the antioxidant already present in the starting polymers (not described in the materials and method section), the overall antioxidant content for Polymer-BE and polymer-BRE would be higher compared to the reference sample.

The FTIR characterization on its own does not add any relevant information to the article if it is not coupled with another characterization technique (e.g. GPC) in order to understand the chemical changes to the polymeric chains.

GPC and/or rheological characterization would add a significant contribution in the understanding of the type of degradation occurring in the samples upon UV exposure, which at the moment it is yet unclear.

The proposed natural antioxidants do not seem to prevent degradation. The alteration of the color is a standard effect of almost all the antioxidant. It is yet unclear if the changes in the mechanical properties are due to the crosslinking of the polymers or thermal degradation. The authors claim that the proposed antioxidant can be used in packaging applications; however the data showed an increase brittle effect which is the main issue in using PLA. PCL is more ductile and, therefore more suitable for packaging, however the mechanical characterization data also show an increase in brittleness and decrease in the thermal properties (i.e. faster degradation curves at the TGA and appearance of a low melting temperature peak).

Below are more specific comments:

Page 2, lines 48-55. This part does not add important information to the manuscript.

Page 2, line 76. AO is defined later. Please move to the first appearance of AO.

Page 3, line 109. BRE is not defined.

Page 3, line 110. FCR is not defined.

Page 3, line 119: DPPH is not defined.

Page 3, line 121. The ability of a radical scavenger is not to provide a hydrogen, but to effectively trap reactive radicals. Radical scavengers are often conjugated structures containing aromatic rings (e.g. polyphenols) to delocalize the reactive radical.

Page 5, line 187. It is well-known that hydrolysis is the main reason for thermal degradation of polyesters. The reviewer appreciate that the authors are discussing the degradation in presence of oxygen; however, due to the presence of humidity (also mentioned by the authors) hydrolysis cannot be excluded.

Page 5, table 2. If Tm1 is defined as the peak melting temperature at ≈38 oC and Tm2 as the peak melting temperature at ≈67 oC, then the Tm values for PCL, PCL-BE, PCL-BRE should be placed under Tm2.

Page 6, figure 3. The baseline of the thermograms is not good at all. The transitions can barely be seen. There is no point in exceeding 240 oC. If you cannot re-measure the sample with a better baseline, please cut to the temperature of 240 oC.

Page 6, figure 4. There is no point in exceeding 240 oC considering that the baseline is not flat anymore. Please cut to the temperature of 240 oC to better appreciate the transitions.

Page 6, lines 206-212. The review disagrees on the conclusion that degradation is slower in the case of PCL-BE-UV and PCL-BRE-UV compared to PCL-UV based mainly on the Tonset where T10 and T50 are lower compared to those of PCL-UV

Page 7, figure 5. The thermogravimetric characterization of the PLA samples reveals a weight loss in the region below the thermal degradation. This is a signature of volatiles present in the PLA or products formed upon exposure to oxygen which might affect the study. The above could be the reason why the baseline for the PLA samples is not flat.

Page 8, lines 224-226. If the main chain degradation is by cross-linking as the authors claim earlier, then the sentence is not correct. In fact, a crosslinked PLA would have higher Young’s modulus.

Page 8, lines 230-231. The authors claim “a slight increase in elongation at break in all AO systems…”; however, the elongation at break in table 4 for PLA-BE and PCL-BE is lower and same respectively.

Page 11, line 288. C-O-C is part of the ester bond, cannot be called ether.

Page 12, lines 297-299. The bands at 2865 and 2945 of PCL are not due to hydroxyl groups, but the symmetric and asymmetric bending vibrations of the methylene unit.

Page 12, lines 299-300. Please revise the sentence. The band correspond to the stretching vibration of the carbonyl of the ester group.

Page 13, line 319 and captions of figures 13-14. The reviewer suggests to clarify that the images G-I are same of D-F but at lower magnification.

Page 13, lines 326-327. SEM is a surface technique and cannot observe the inner part of the material. The significant loss in the elongation at break and the change in the other mechanical properties suggest that the degradation is not only superficial, but involve the bulk material.

Author Response

Dear reviewer,

Reviewer 2 Report

The study investigated effect of an antioxidant based on red beetroot extract on the abiotic stability of polylactide and polycaprolactone, and conducted a series of experiments such as characterization of BRE, thermal analysis, mechanical properties, colour measurement, FT-IR, SEM. In my views, this manuscript can be accepted after minor revision. Below are some points that I would like to bring to the authors' attention:

  1. Please adjust the layout of the article, especially the diagrams and corresponding topics, to make it more beautiful.
  2. For Table 1, Table 2, Table 3, you should indicate the meaning of Tg, Tm.
  3. In this study, why did the author choose bentonite as a carrier?
  4. It is recommended to add the FTIR of BE and BRE.
  5. What are the advantages of BRE as an antioxidant?

Author Response

Dear reviewer,

Round 2

Reviewer 1 Report

2) If the FT-IR study was performed solely to test possible structural changes once mixed with the AO; then what is the point of studying the effect of the UV irradiation? A significant discussion on the effect of the UV irradiation is present.

Page 5, table 2. The Tm1 and TM2 values for PCL-UV (all) are wrong. Please correct. Moreover, the values of enthalpies should be rounded up to the units; their determination is not accurate to the decimal point.

Page 8, lines 227-232. The second sentence is basically a repetition of the first without the link to crosslinking. Please delete the second sentence.

Page 11, line 297. Correct eester with ester. Moreover, C-O-C is not the ester group, but it is part of the ester group (C(=O)-O-C).

Page 12, line 309. Space is missing between “theester”

Please review all the citations for missing page numbers, DOI, bold for years, capital words

Author Response

Dear reviewer,

Reviewer 2 Report

accept

Author Response

Dear reviewer,

thank you for your revision which helped us to improve our manuscript.

We wish you all the best in your research.

Yours sincerely, Team of authors